# CO-Releasing Materials: An Emphasis on Therapeutic Implications, as Release and Subsequent Cytotoxicity Are the Part of Therapy

**DOI:** 10.3390/ma12101643

**Published:** 2019-05-20

**Authors:** Muhammad Faizan, Niaz Muhammad, Kifayat Ullah Khan Niazi, Yongxia Hu, Yanyan Wang, Ya Wu, Huaming Sun, Ruixia Liu, Wensheng Dong, Weiqiang Zhang, Ziwei Gao

**Affiliations:** 1Key Laboratory of Applied Surface and Colloid Chemistry MOE, School of Chemistry and Chemical Engineering, Shaanxi Normal University, Xi’an 710062, China; faizanattari@snnu.edu.cn (M.F.); huyongxia@snnu.edu.cn (Y.H.); yyw@snnu.edu.cn (Y.W.); wuya@snnu.edu.cn (Y.W.); hmsun@snnu.edu.cn (H.S.); wsdong@snnu.edu.cn (W.D.); 2Department of Biochemistry, College of Life Sciences, Shaanxi Normal University, Xi’an 710062, China; niazpk@hotmail.com; 3School of Materials Science and Engineering, Xi’an Jiaotong University, Xi’an 710049, China; niazikifayat@stu.xjtu.edu.cn; 4Institute of Process Engineering, Chinese Academy of Science, Beijing 100190, China; rxliu@ipe.ac.cn

**Keywords:** CO administration, therapeutic agent, pharmaceutical drugs, heme oxygenase, CO-releasing materials, CO-releasing molecules, organometallic complexes, pharmacokinetic functions, pathological role, CO kinetic profile, cellular targets

## Abstract

The CO-releasing materials (CORMats) are used as substances for producing CO molecules for therapeutic purposes. Carbon monoxide (CO) imparts toxic effects to biological organisms at higher concentration. If this characteristic is utilized in a controlled manner, it can act as a cell-signaling agent for important pathological and pharmacokinetic functions; hence offering many new applications and treatments. Recently, research on therapeutic applications using the CO treatment has gained much attention due to its nontoxic nature, and its injection into the human body using several conjugate systems. Mainly, there are two types of CO insertion techniques into the human body, i.e., direct and indirect CO insertion. Indirect CO insertion offers an advantage of avoiding toxicity as compared to direct CO insertion. For the indirect CO inhalation method, developers are facing certain problems, such as its inability to achieve the specific cellular targets and how to control the dosage of CO. To address these issues, researchers have adopted alternative strategies regarded as CO-releasing molecules (CORMs). CO is covalently attached with metal carbonyl complexes (MCCs), which generate various CORMs such as CORM-1, CORM-2, CORM-3, ALF492, CORM-A1 and ALF186. When these molecules are inserted into the human body, CO is released from these compounds at a controlled rate under certain conditions or/and triggers. Such reactions are helpful in achieving cellular level targets with a controlled release of the CO amount. However on the other hand, CORMs also produce a metal residue (termed as i-CORMs) upon degradation that can initiate harmful toxic activity inside the body. To improve the performance of the CO precursor with the restricted development of i-CORMs, several new CORMats have been developed such as micellization, peptide, vitamins, MOFs, polymerization, nanoparticles, protein, metallodendrimer, nanosheet and nanodiamond, etc. In this review article, we shall describe modern ways of CO administration; focusing primarily on exclusive features of CORM’s tissue accumulations and their toxicities. This report also elaborates on the kinetic profile of the CO gas. The comprehension of developmental phases of CORMats shall be useful for exploring the ideal CO therapeutic drugs in the future of medical sciences.

## 1. Introduction

Carbon monoxide (CO) is considered harmful due to its toxic behavior since the last century. It has a tasteless, odorless and colorless nature. Its colorless nature allows CO to remain undetectable even at high concentrations and toxic levels, thus marked as the “silent killer” [1,2]. This poisonous CO behavior is exerted due to the formation of carboxy hemoglobin (COHb) along with oxygen present in the mainstream blood circulation. Haldane and Douglas scientifically explored it the first time through dissociation curves of CO-hemoglobin using a constant percentage of CO along with a variable percentage of oxygen at an atmospheric pressure [3,4]. The ubiquitous enzyme, heme oxygenase (HO) has been investigated in most of the biological species. In the middle of the 19th century, two scientists Tenhunen and Schmidt discovered the intracellular CO production by heme oxygenase with an enzyme being the heme catalyst [5,6]. HO is categorized under two isoforms: HO-1 (HMOX1; gene name), with its capability to remain inducible in all cell functions; and HO-2 (HMOX2; gene name), that is constitutively expressed and substantially contained in vasculature and testes [7,8]. HO-1 is identified as an element exclusively found in spleen and liver [9]; however it might be influenced by varying intensity in most biological tissues. Both HO-1 and HO-2 indicate the rate-determining step, drag-out biliverdin from the heme conversion with the CO release and Iron product associated with a tetra pyrrole ring. Biliverdin using biliverdin reductase transforms into bilirubin while generating ferritin quickly from the Iron segregate (Figure 1) [10]. The released amount of CO attaches with the Iron containing objects due to its higher diffusion rate and tendency. It tends to make itself bonded with blood in the circulatory system; and ultimately it is exhaled through lungs. CO causes a common sagginess by bringing affliction for the mammalians, completely dependent on oxygen for the blood transport system and mitochondrial respiration. Collectively, endogenously generated CO is featured in the physiological role. Generally, a low dose of CO gas endures tremendous benefits and can achieve remarkable therapeutic targets.

CO is endogenously produced by either specific enzymes or through gas transmission into a biological system, both types exhibit the physiology and pathophysiology functions through inter- and intra- cellular interactions. The endogenously produced CO also raises their potential as a therapeutic agent. The scientists and researchers are availing this opportunity and spending their time and energies for developing modern drug techniques [11]. Their aim is to explore the modern work with the novelty of this great strategy.

### 1.1. CO Biological Scope

The CO gas is known for its leading role as a molecular messenger in the physiological process for the nervous system [12] and also for following some important therapeutic treatments [2,13]. It has the potential for anti-inflammatory [14], anti-proliferative [15], anti-atherogenic [16], anti-allodynia [17], anti-nociceptive [18], anti-hyperalgesia [17], and anti-apoptotic [19] effects. It is vital for vasodilatory phenomena reducing intraocular pressure [20], immunosuppressive administrated medications [21] and also has the capability to develop the pathological cellular process (Figure 2) [22]. CO also has many advantages for different biological organs: Organ transplantation [2], protection [23] and preservation; heart [24,25,26]; kidney [23,27,28,29]; liver [30,31]; lungs [32,33]; pancreatic islet [34] and the small intestine [35]. It is helpful to de-escalate the Ischemia/Reperfusion Injury (IRI) [36], mitigate the myocardial infarction and allograft rejection [37], stimulate the cytoprotective [38], and is also involved in anti-microbial [39] and anti-hypertensive activities [40]. It has a modulated utility for heme-dependent proteins like mitochondrial cytochromes and NADPH [28,41]. Moreover, the intercellular CO production by heme oxygenase has proved itself as a valuable reagent [42]. The pharmaceutical dose through endogenous CO enrichment or exogenous direct transformation is flourishing and will be attracted as a therapeutic interest lately.

### 1.2. CO Therapeutic Ways

Mainly, there are two ways to insert the CO molecules as a therapeutic agent inside the human body, i.e., direct and indirect CO insertion. The direct inhalation has not been preferred, owing to its rise in the COHb level above 10% and lack of tissue selectivity (Figure 3). Moreover, it provides a direct interaction of the CO and lungs only while detainment of CO is also observed in this method. These limitations don’t allow CO to approach other biological organisms for therapy. To overcome these problems, researchers have developed an alternate strategy called “Exogenous Endeavor” for obtaining the required therapeutic actions. In the early 19th century, researchers also recognized the toxic gas NO as the nitro drug having therapeutic impacts. The nitric medications are well demonstrated as nitric oxide-releasing molecules (NORMs), and that established their well reputation afterward CORMs cogitation [43].

### 1.3. Why Exogenous Endeavor is Required?

The CO-releasing fragment is basically an exogenous endeavor that has opened up the paths for therapeutic treatments (Figure 4). The exogenous stakeholder CO, makes space for searching the affected sites, reaches at the diseased tissue site and makes conflict/collusion with the selected tissues for the destruction of damaged organs or/and diseased cells. If required, the CO-releasing rate can be regulated and modified according to specs. To disintegrate CORMs and CORMats into CO and metal residue, numerous activators are being administratively applied for controlling the CO liberation rate that has already been experienced in Photo-CORMs and Photo-CORMats through Ultra Violet (UV), Visible and NIR light with on/off switching facility [11,46,47,48,49,50]. The photon energy also has a utility to extract CO from its parent organometallic ligand. The main advantage of CO’s exogenous interactions with the mammalian organism is that it reduces the CO moiety to be directly induced into blood streams for maintaining the COHb under allowable serum levels (up to 10%). Without an endogenous CO administration, it is quite challenging to get productive outcomes.

### 1.4. Clinical Translations

In spite of its hazardous nature, a controlled CO direct inhalation has some therapeutic benefits as well. A clinical trial of controlled CO dosage was conducted on healthy volunteers for temporary paralysis of intestines known as Post-Operative Ileus (POI), and usually every patient is engaged in this POI after surgery of the abdomen. This clinical study revealed that serious POI complications could be significantly reduced if the CO dose (~250 ppm) is inhaled before and after the colon surgery (ClinicalTrials.gov identifier: NCT01050712). Another CO clinical translation test also shows a transplantation protection when the CO–saturated medium is provided for harvesting islets as it protects the cell from chronic pancreatitis (ClinicalTrials.gov identifier: NCT02567240).

This valuable intensive information about the CO therapeutic analysis plays a vital role for all the researchers’ attention. Although there are few unfavorable emblems associated with releasing CO moiety, but recent outcomes of therapeutic potential helps to promote CO as preclinical stems [13]. This novel idea was initiated through clinical and pre-clinical trials for either the direct inhaled therapy [13] or oral intake of CO-releasing substances including CORMs or CORMats, which is a modern result of the professional chemistry enterprise [51].

### 1.5. Challenges and Demanding Features of CORMs and CORMats

Although CORMs and CORMats have a tremendous therapeutic utility but it also possesses some sort of following limitations for releasing the embedded CO.
Availability, solubility and stability of reagents under ambient conditions.Feasibility to release the captured CO from in situ CORM.Controllability to release CO kinetics up to a desired level.Prone to toxicity which arises due to the transition metal foundation of metal-ligand fragments.

The abovementioned fundamentals have been discussed as a prescribed domain while exploring the CO discharge. Considering the CO gas as a therapy treatment based on CORMs and CORMats are easier to control by the transportation of gas molecules rather than the direct CO gas intake. Moreover, the rapid diffusion of these small molecules limits their ability to concentrate in specific tissues. Many challenges also arise during movability of the CO gas molecules by these strategies. Both the carbonyl transition metal and all its degradation products are biologically toxic in nature. Hence, it is difficult to manage the CO discharge with respective biological tissues. Particularly, the release of the CO molecules from CORMs also participates in depositing heavy metal ions inside the human body, which could be harmful for biological organisms.

### 1.6. Triggers

There are different ways to disintegrate CORMs/CORMats for the release of CO moiety by means of; peculiar physiological conditions [52], trigger by temperature [53], activation by an enzyme [54], pH alteration and increase in reactive oxygen species (ROS) concentration, accessibility to distinct wavelength of light [55,56], either using thermal degradation or ligand exchange/substitution or both [53] and prototypically activation through oxidation mechanism [11,57,58]. In order to deliver the CO molecules at a specified therapeutic rate, it is necessary to characterize the CORMs and CORMats entities, as quantification of such mechanisms might be contingent with its physical conditions like O_2_, temperature, assay solution and light conditions. 

### 1.7. CO Identifier

The authorized CO identification method consists of the following techniques: Electrochemical assays [59,60], laser infrared absorption [61] and gas chromatography [62]. Moreover, a colorimetric CO sensor facility is an alternative platform, especially for observing the CO behavior in living cellular tissues or/and organs [63,64]. The vibrational spectroscopy technique (such as IR, infrared and Raman) is one of the quickest ways to monitor the CO attached with transition metal through carbonylation at distinctive bond ranges. To date, the plethora method “Myoglobin Assay” famous as “*Gold Standard*” has been in operation for in vitro interrogation of the CO release from CORMs and CORMats. The “Myoglobin assay” is also a worthy and standard method for observing the behavior of the CO release kinetics in a biological environment and monitors the performed activity of CO (Equation (1)).
Mb + CO → Mb-CO(1)

Recently, It has been observed that binuclear Rhodium compounds, i.e., *cis*-[Rh_2_(C_6_H_4_PPh_2_)_2_(O_2_CCH_3_)_2_](HAc)_2_] can detect CO with a substantial selectivity and superior sensitivity [65]. The gas-phase IR spectroscopy is the most reliable and high-resolution technique for analyzing the CO release activity. For directly sensing the CO, various gas chromatography (GC) detectors have been introduced to date including the gas chromatography-mass spectrometer (GC-MS) [66], reduction gas detector (GC-RGD) [67] and thermal conductivity detector (GC-TCD). The fluorescent probe has the ability to recognize the CO-release entity even at a low concentration as compared with the myoglobin assay, but it is unable to operate in a short interval kinetic measurement (for that at least one hour interaction is required) [68,69]. 

### 1.8. The Development Phases of CORMs Motifs

#### 1.8.1. Metal Carbonyl Complexes (MCCs)

To construct the bonding relation between the CO and low valent metal ions for producing carbonyl complexes (M-CO), the M-CO bonds must undergo an inert ambiance along with reducing conditions, which are mostly feasible in organic solvents. Irrespective of a metal physical state, the CO gas can react and develop the volatile metal carbonyl complexes (MCCs) for example, Ni(CO)_4_ and Fe(CO)_5_ [70]. MCC acts as a core entity for organometallic transition chemistry. The general representation for MCCs is [M_m_(CO)_x_L_y_]^z±^[Q^±^]^z^ [53], in which, M, L are the basic entity known as transition metal (B, Cr, Mn, Fe, Co, Mo, Ru, Rh, W, Re, Ir) [71], and ancillary ligand might be the C, O, P, N, S or halide ligand. Furthermore, Q and z represent the counter ion and overall complex charges. If no counter ion is available, then z will be zero. Moreover, m, y, z are calibrated as stoichiometric coefficients and x and m values should be ≥ 1 [53]. Modern and classic complexes can be distinguished by two determining factors: Low oxidation state (OS) or either very low oxidation state (LOS) and total valance electron occupied in the outer coordination sphere. For 4th, 9th, and 10th groups, the compounds were observed to have 16e^-^ configuration and the rest of the complexes were generally observed having 18e^-^ configuration. Each of these commodities must comply with the chemical, biological and physical characteristics of MCCs. Furthermore, it needs to be precisely selected in the configuration of pharmaceutical CORMs [71].

It is important to note that CORM and CORMats are stable in the aqueous medium, and it is also feasible to store it under ambient environmental conditions like the majority of other pharmaceutical drugs. Their circulation must be ensured, as it needs contact both with the diseased and damaged tissues. Moreover, the potent and non-toxic metabolites may be left behind after the CO removal. This is exactly what is required and regarded as therapeutic features. It sets the basic pattern, in the line of action for the development of such challenge-able MCCs. Mostly, during administration MCCs incorporates with the organic solvent and results in traditional oxygen-free atmosphere. These medical conditions might be different from a variety of other biological surroundings, considering that most reactants and their resulting complexes are uncertain under ambient conditions, i.e., oxygen and humidity [71]. Hypothetically, this biological activity of MCCs remains toxic in nature like Ni(CO)_4_, and (MeCp)Mn(CO)_3_ (MMT) as an anti-knock gasoline additive [72]. Hence, the common MCCs chemistry can act as a simple guideline for the development of pharmaceutical CORMs. Currently, this research is focused on novel strategies for establishing MCCs-CORM’s activity and specifically for therapeutic purposes.

The above discussion suggested that MCCs when triggered as CORM’s, become a competitor for the CO availability during its decomposition. In organometallic complexes the releasing strategy is as follows: A new incoming ligand (L’) can push itself to a metal center resulting in a new bond, which establishes and influences on the coordination number. The elevated coordination number then promotes it to elongate the M-CO bonds, and eventually it then breaks. Consequently, the CO is liberated by this method and the new L’-M bond is constructed. This information provides the foundations of the CORMs concept. The chemistry of MCCs provides per se different strategies and has an adverse impact on the CO release (Scheme 1) [53].

#### 1.8.2. Proposed Strategies for CORMs Development 

CORMs based on metal-to-ligand charge transfers (MLCT) morphology. These CORMs are the elementary motifs with organometallic ingredients; corresponding to a series of MCCs occupied at the transition metal core. Other exclusive features are mentioned below: Structural variance of unique chemistry;Expected divergence with different oxidation states;Covalently bound with the metal center;Assisting alterations for the attached carbonyl ligands;The dynamics of co-ligands binding;Tendency of the outer coordination sphere.

As abovementioned, the spectroscopic nature of MCCs confirms the identification and recognition of significant trace elements like ruthenium [73,74], manganese [49,74,75,76,77,78], iron [37,79], cobalt [80,81], tungsten [82], osmium [83], molybdenum [82] and rhenium [84]. The developed organometallic carbonyl complexes CORMs are CORM-1, CORM-2, CORM-3, CORM-401, ALF492, CORM-A1, B_12_-ReCORM-2, Re-CORM-1, CORMA-1-PLA and ALF186.

Along with organometallic complexes, the miscellaneous compounds can be nominated as the CORMs family. In this scenario, numerous nonmetallic compounds [85] have been accomplished by entertaining the CO release such as silica-carboxylates [86], borano-carbonates [87], borano-carbamates [88], xanthene carboxylic acid (XCA) [89], unsaturated cyclic diketones (DKs) [90], methylene chloride (MC) [91,92], meso-carboxy BODIPYs [46], hydroxy-flavones [93] (Scheme 2). Furthermore, 1,2-disubstituted ferrocenes belongs to an aldehyde family, unfavorably elicits the toxicity and its slow release mechanism restricts the researchers from developing another nonmetallic CORMs (NCORMs). The main drawbacks of NCORM are potentially a low CO content releasing, and always producing organic molecules along with the CO moiety. Anyhow, NCORM clinical traits have shown their utility to communicate with biological activity [94]. 

### 1.9. CORM’s Therapeutic Scope

Strategically, the synthesis route of CORMs development is not the only objective. The main theme of this CORM’s innovation is to obtain the therapeutic advantage eventually. The biological significance of CORM is associated with their bacterial performance in cells lines, (i.e., standard myoglobin assay). The important biological roles of CORMs are listed below (Table 1). 

### 1.10. Solubility

For making CORM acceptable as a pharmaceutical drug in the mammalian biological system, solubility is one of the most prominent factors where the researcher can evaluate the proficiency of the product. Solubility estimates how much CORMs and CORMats are convenient for the practical demonstration. CORM-2 is soluble in DMSO, olive oil and PEG [95,112], while CORM-3 contributes to the water compatibility with a weak acidic nature (pH = 3) [26,37]. Specifically, CORM-A1 possesses the water solubility and stability but it breaks-down immediately after liberating the CO under acidic condition (pH = 11) [87]. CORM-ALF186 can afford the disintegration in the water system and is unstable at an aerobic condition [113] and ALF062 is soluble in methanol and DMSO, while it remains unstable in the air [98,113]. Furthermore, CORM-1 has the compatibility with DMSO and ethanol [114]. 

Although the CORMs motif is good for releasing the CO moiety, but the tissue selectivity and targeting sites dilemma has reduced its overall biological performance and hence lost its therapeutic significance. Most importantly, the toxicity of organometallic complexes is handled very poorly, so to reduce the toxicity and increase its reactivity, it requires an exploration of all the alternative strategies. Therefore, the researchers have moved from CORMs to CORMats.

## 2. Research on New CO Transport Materials 

As discussed, the MCCs are the admissible and professional class of (soluble) CORMs; however, it has been imperative to examine their probable shortcomings. In fact, a small number of organometallic compounds can be manipulated for pharmaceutical agents predominantly caused by the side-reaction of metals with biological chemical compounds, (e.g., nucleophilic or even electrophilic side chains of proteins) together with the toxicity of several heavy metals. Water-soluble CORMs are approaching the entire body organism and it could accelerate the toxicity against healthy tissues. The spatial and acceptable releasing rate of it into biological tissues/cells is still the utmost challenge. Furthermore, the CO-releasing activity inevitably accumulates a metal and co-ligand fragment, probably takes part in the biological activity as well. This residue (i-CORMs) can be managed through the insoluble framework. On the basis of abovementioned issues and challenges, new and compatible CO transport materials and strategies are emerging in order to get rid of the CO lethal gas dilemma and to convert it into a valuable clinical agent. CORM-1 [115], CORM-2 [116,117], CORM-3 [118] and CORM-A1 [87] have been tested in various disease models to observe their therapeutic effects and to obtain its surprisingly outcomes in typical clinical conditions [2,119]. CORM-3 has good cure-ability for inflammatory disorders like rheumatoid arthritis, osteoarthritis and collagen-induced arthritis (CIA) [97,120,121]. CORM-A1 provides ameliorated course in experimental auto-immune uveoretinitis (EAU) [122], while CORM-2 attenuates the tumor proliferation [123] and a considerable enhancement the coagulation and slow-down of the fibrinolytic bleeding [117,124] and improves survival in the liver injury affected by cecal ligation and perforation (CLP) [116,125]. CORM-3, CORM-2 and ALF-062 corroborates with antimicrobial functions [98,126,127,128,129,130,131]. The CO is encouraging the proliferation of endothelial cells, progenitor cells and regulatory T-cells [34,132,133]. There is still more interrogation required for further improvement to employ practical knowledge. So, the development of the solid CO precursor in tandem with peculiar trigger for releasing the enclosed CO gas commodity is an imperative research motive. To date, due to the unavailability of a safe delivery system for CORM; none of those prescribed formulations could have been employed in humans as a direct dose for respective damaged tissues or disease. Although a few scientific proposals have been presented in that scenario for making CORMs as a clinically viable project, but none of them exhibits the secure transportation material system for the patient’s right choice. All that discussion pursued that the nanoscale and macromolecular carrier system could be exploited to obtain a selected tissue enrichment and proposed mechanism strategy for CORMs delivery (Scheme 3) [134,135]. 

The clinical trials on CORMs proved that CORMs exhibited the important biological applications, but after CORMs, the degradation metal residue (i-CORMs) also caused toxicity unfortunately [136]. The prohibited i-CORMs activity containment is a big challenge for the researchers. To reduce toxicity and capture the i-CORMs toxic moiety, scientists have explored a strategy known as CORMats. In this strategy, firstly, CO is entrapped inside the CORMats through specific administration, and then upon certain conditions the captured CO is escaped out. Several scaffolds and conjugated formulations have been introduced in this scope and it is still under investigation by compatible conjugate CORM’s such as Ruthenium-MCC (Ru-MCC) and Manganese-MCCs (Mn-MCC) by different nano-transporting services such as Iron MOFs [137,138], peptide [139,140,141,142,143,144], micellization [55,59,145], protein [121,146,147,148,149], vitamins [150,151,152,153], co-polymer systems [47,154,155,156], nanofiber gel [142], inorganic hybrid scaffolds [157,158,159,160] and metallodendrimers [161] (Figure 5). The intrinsic toxicity control of i-CORMats is the top priority for each developed system.

### 2.1. Micellization

Hubbell et al. engineered the micellization technique as the CO-producer with reduced diffusion; creditably targeted to the distal tissue draining sites [59]. Micelles were synthesized by the tri-block copolymer composed of poly(ornithine acrylamide) block and poly(ethylene glycol) block (hydrophilic nature) hosted by [Ru(CO)_3_Cl-(ornithinate)] moieties with poly(n-butylacrylamide) block (hydrophobic nature). The CO-releasing micelles consists of a triblock copolymers (Figure 6): A hydrophilic poly(ethylene glycol) (PEG) fragment that stabilizes the micelles; a poly-OrnRu fragment that releases CO. A hydrophobic poly(n-butylacrylamide) fragment drives to construct the micelles forms.

The micelles polymer can be used as a pharmaceutically acceptable carrier to solubilize the poorly soluble drugs and produces the therapeutic effect against the targeting sites. Probably it promotes the reduction in toxic effects of the drugs on normal tissues and organs. Significantly, the toxicity of the Ru(CO)_3_Cl moiety is well-reduced in the polymer micelle due to the stealth characteristic of the PEG fragment. Moreover, the micelles moderately respond to human monocytes against the lipopolysaccharide (LPS) -induced inflammatory disease model. Importantly, poly(ethylene glycol) attenuates the toxic feature of [Ru(CO)_3_Cl(amino acidate)] moieties. The addition of cysteine allows the release of CO from an occupied area with a slower rate as compared to [Ru(CO)_3_Cl(glycinate)] (CORM-3). The release of CO from micelles was tested in a myoglobin assay and it has been found to be slower than CORM-3. The diffusion of the Orn-Ru substrate is facing hindrance in the cells due to the micelles stereoscopic effect. Anyhow, the mechanism approach of CO-releasing is not obvious. It has been evidently proved through experiments that thiol compounds such as cysteine, glutathione and protein are compatible to induce the CO release from micelles.

Hiroshi Maeda et al. incorporated the tricarbonyldichlororuthenium dimer (CORM-2: [RuCl(μ-Cl)(CO)_3_]_2_) as water-soluble styrene-maleic acid and copolymer (SMA) while gaining the optimum half-life and numerous therapeutic effects [145]. They established the micellization structure for encapsulating the CORM-2 (SMA/CORM-2) (Figure 7). The micellization has good water solubility and it is compatible with the aqueous environment. The sustain CO kinetic profile performs well in vivo bioactivity such as murine model of inflammatory colitis. The half-life of this complex was almost 35-folds compared with the free CORM-2.

In spite of the ligand exchange CO-release mechanism, photo light is able to disintegrate the CORMats moiety as the CO donor. Robert Igarashi and Yi Liao explored the micelle-based photo-CORMs synthesized by the cyclic α-diketones (α-DK) encapsulation [55]. These CORMs require visible photo light for releasing the embedded CO. This research demonstrates the therapeutic potential of CORMs. The photo-activated micelle CORMs strategy has been explained in Figure 8. During the study of these micelle CORMats on the cell proliferation, it has been found that no difference was monitored in the viability of cells in response to the micelles of DKs.

### 2.2. Peptide

John reported a self-assembled amphiphilic peptide (PA) that was used to produce the CO [142]. Thereby, a covalent combination of a hydrophobic alkyl chain and a hydrophilic short sequence peptide endures the self-assembled peptide chain material. Amphiphilic peptides can spontaneously release CO and are prone to toxicity themselves. In that perspective they first designed the amphiphilic peptide PA1; which contains a β-aspartate residue to generate the NH_2_-CH-RCOOH unit closely resembling the CORM-3 fragment. Next, PA1 and [Ru(CO)_3_Cl_2_]_2_ were synthesized in the presence of sodium methoxide at room temperature to synthesize the CO-releasing peptide PA2 (Figure 9). The CO kinetic release curve proved that peptide P2 was synthesized in an aqueous solution like the first-order rate constant of CORM-3. The half-life of the CO released from both sources is quite identical. The half-life of CO released from CORM-3 is 2.14 ± 0.17 min, while the half-life of the CO released peptide P2 is 2.16 ± 0.05 min. In order to increase the half-life of CO released from P2, the incorporation of nanofiber gel PA2 and a strong gel PA was made. The half-life of the CO released from this nanofiber gel was significantly increased (~17.8 min) compared to PA2 and CORM-3 in an aqueous solution.

Rather than encapsulating a mere CO segment inside the transport materials, the CORMs entity incorporates with different functional groups of parent CORM’s ligand commodity. In that scenario, peptide is linked with the Manganese-based Photo-CORM [Mn(CO)_3_]^+^ ligand tpm (tris(pyrazolyl)methane) using a Pd-catalyzed based Songashira cross-coupling mechanism and click reaction at N-terminal (azide-) and side-chain (iodoarene-) functionalization [140]. 

Ulrich Schatzschneider exposed the peptide linkage for the CORMats development [144]. They introduced the Mo-carbonyl [Mo(CO)_4_(bpy^CH3,CHO^)] associated with aldehyde functional groups at the peripheral position. The bioactive β-target peptide ligand 2,20-bipyridine (bpy) attached with molybdenum-carbonyl by *N*-terminal bonds of aminoxy acetic through catalyst-free and bio-orthogonal oxime ligation (Figure 10). The photo-activated CORMats gets activated upon 468 nm photons lights irradiations.

Radacki and Ulrich Schatzschneider jointly synthesized the Manganese carbonyl complexes [Mn(bpea^NHCH2C6H4CHO^)(CO)_3_]PF_6_ [139]. The peptides ligand 2,2-bis(pyrazolyl)ethylamine (bpea) is bearing aminoxy, azide and N-terminal alkyne residues (Figure 11). The researchers applied the transforming growth factor β-recognizing (TGF-β) peptide sequence for developing the photo-activated delivery agent. This peptide conjugation could be utilized for further development of new CORMats.

The JJ Kodanko group successfully synthesized the ionic water-soluble compound [Fe^II^(CO)(N4Py)] by a method of continuous CO bubbling through a ligand N4Py and one equivalent of Fe^II^(ClO_4_)_2_ under the action of the organic solvent acetone (ClO_4_)_2_ (Figure 12A) [141]. A myoglobin experiment shows that the compound is stable under dark conditions and its releasing half-life is more than one day. When it was irradiated with 365 nm ultraviolet light, the CO can be quickly released. According to MTT experimental studies, [Fe^II^(CO)(N4Py)](ClO_4_)_2_ exhibited the effective cytotoxicity against human prostate cancer cell line (PC-3) under light-induced conditions. When the concentration reaches 10 μM the cell survival rate was monitored as 63% of the control group. In order to further investigate the CO release behavior, the carbonyl segment with acetonitrile was replaced. The UV-vis analysis found that the substitution process is very slow [Fe^II^-(MeCN)(N4Py)]^2+^ and the concentration of acetonitrile was a quick step to replace CO. Moreover, it was also found that the N4Py ligand can be modified with a peptide (Ac-Ala-Gly-OBn) to obtain a peptide-conjugated photo-induced release molecule (Figure 12B). Biological experiments have showed that the peptide chain conjugation might be evaluated for the improvement of cell-specific itself or tissue-specific CO transport properties.

In the development of Photo-CORMats, metal-coligand plays a vital role in the photoexcitation at a prescribed wavelength. This allows the photons energy to penetrate and push the CO molecules to pull out these molecules from the metal-ligand fragment. In other words, these wavelengths are providing extra energy that enables the CO molecules excitation from its parent location. The CORM-2 and CORM-3 are hydrolytically active. Synthetically, these CORMs could be transformed into the photo-activated reagent. Leone Spiccia and Ulrich Schatzschneider described the ruthenium (II) dicarbonyl complexes functionalized with 2-(2-pyridyl)pyrimidine-4-carboxylic acid (CppH) (Figure 13A) [143]. They were able to successfully construct the monomeric PNA backbone with ruthenium (II) di-carbonyl complexes to produce ruthenium (II) dicarbonyl dichloride-based PNA-like monomer [RuCl_2_(Cpp-L-PNA)CO_2_] (where PNA= peptide nucleic acid) (Figure 13B).

### 2.3. Proteins

G. J. Bernardes et al. disclosed that CORMs was compatible with the protein complexes transportation (Figure 14) [121]. They presented that the Ru^II^(CO)_2_-protein complexes using the reaction between CORM-3 and histidine fragment at the protein surface, the spontaneous CO is released to deliver in cells and mice. They also discussed that plasma protein acts as a CO carrier for in vivo by the CORM-3 formulation. Therapeutically, the controlled CO release favors in downregulation of the cytokines interleukin, i.e., IL-6 and IL-10.

Another Ruthenium (II) carbonyl reagent *cis*-[Ru(CO)_2_(H_2_O)_4_]^2+^ has been reported for the spontaneous CO release in live cells using histidine (His) metalloprotein and retained at IL-8 (Figure 15a) [148]. The *cis*-[Ru(CO)_2_]^2+^ carbonyl segment could be produced by aqua dirutheniumcarbonyl *cis*-[Ru(CO)_2_(H_2_O)_4_]^2+^ (Figure 15b). It was also explained that metalloproteins can be modified as organometallic pro-drugs rather than catalysis. Such artificial metallohydrolase performance can be compared with the human carbonic anhydrase (CA)-II.

In spite of the CORMs fragment incorporation with protein, Takafumi Ueno et al. explored the cages of protein for CO releasing [149]. They administrated the ferritin (Fr) cage of protein for capturing the CORM-3 moiety (Figure 16). Furthermore, it was observed that the half-life of the CO release could be enhanced; which indicates a good sign for ideal drug development. When they interrogated their performance at the biological sites, they described that the nuclear factor kappa B (NF-κB) becomes 10-times higher than the parent CORM-3. The CORM-3 protein cage is quite a unique way of CORM’s engineering.

Additionally, Takafumi Ueno et al. explored the immobilization of the crosslinked hen egg white lysozyme CL-HEWL crystal deposit on MCCs for therapeutic purposes [146]. The scientist disclosed that NF-κB is remarkably high in order to respond to the pathological signals. The extra scaffold Ruthenium carbonyl moiety (Ru·CL-HEWL), was used to induce the NF-κB activation and immobilized Ruthenium carbonyl [*cis*-Ru(CO)_2_X_4_]^2-^ moieties inside the protein cage (Ru·CL-HEWL). Optimist approaches of this transport service bear the potential of the artificial extracellular scaffold.

NF-κB can be regulated by the protein fragment. Susumu Kitagawa et al. disclosed the crystalline assembly of protein with CORM-2 in polyhedra crystals (PhC) [147]. They introduced the ruthenium carbonyls immobilized on hexahistidine. The activation of NF-κB was significantly improved up to six folds. This therapeutic research will lead to further investigation on the extracellular scaffold.

### 2.4. Vitamins

Anna Yu. Bogdanovab et al. presented the cyanocobalamin (B_12_) as a biocompatible B_12_-Re^II^(CO)_2_ scaffold for CORMats (Figure 17) [150]. They incorporated the *cis*-*trans*-[Re^II^(CO)_2_Br_2_]^0^ core having 17e^-^ dicarbonyl complexes. This research also elaborated that ReCORM-1 is compatible with B_12_ for pharmaceutical applications, and the obtained cobalamin conjugates are also feasible with aqueous aerobic media. Interestingly, after CO releasing, metal degradation is not involved in toxicity due to the exclusive configuration and metal oxidation of ReO_4_^-^ generation.

The most promising anticancer drug agent is a macromolecular conjugate. The HO-1 and transcription factor Nrf2 are the prime parameters to provide resistance against inflammation and oxidative stress disease. In this analogy, biliverdin is the enzymatic activity of HO-1, while CO directs the therapeutic exploitation. Roberta Foresti et al. found that hybrid molecules were simultaneously involved in the CO liberation and Nrf2 activation [151]. The newly developed CORMats termed as hybrid-CORMats (HY-CORMats) is shown in Figure 18.

After synthesizing the HY-CORMats, researchers further interrogated the biological activities and described the HO-1 expression along with the nuclear accumulation of Nrf2 and showed viability in different cell types (Scheme 4).

A. Pamplona’s group coordinated the galactose with a central metal to synthesize a polyhydric-containing water-soluble CORMats [RuCl_2_-thiogalacto-pyranoside(CO)_3_] (ALF492) through Sulphur bond (Figure 19) [152]. The Sulphur bond coordination of the galactose ligand with the central metal increases the water solubility and biocompatibility. This compound also exhibited the appropriate drug-like properties. The presence of the galactose ligand increases the specificity of liver glycoprotein. A myoglobin study was used to monitor the CO release kinetic profile. The target selectivity of ALF492 can be well administrated. Significantly, when ALF492 is mixed with the antimalarial drug artesunate, ALF492 responds to the effective adjuvant treatment for cerebral malaria. Collectively, this marks the outstanding potential of ALF492 in the treatment of falciparum malaria.

The CC Romao group synthesized the molybdenum-based water-soluble release molecule Mo(CO)_3_(CNCR’R"CO_2_H)_3_ (R’=R"=H, ALF795) (R’=R"=CH_3_, ALF794) (Figure 20) [153]. Myoglobin experiments have shown that compounds were stable and did not decompose in an aerobic aqueous solution for at least 1 hr. ALF794 is less toxic and suitable for drug-like properties. It can deliver CO to acetamido phenol-induced liver in mice with acute liver failure. After 5 min, intravenous injection, the ratio of ALF794 in liver/blood and liver/kidney were reported at 5.27 and 12.58, respectively and ALF795 liver/blood and liver/kidney ratios were observed at 0.33 and 0.50, respectively.

### 2.5. Polymers

The aforementioned most promising anticancer drug agent is a macromolecular conjugate. Ruth Duncan et al. reported for the first time a polymer-anticancer conjugate [154]. These macromolecular drugs consist of at least three (3) parts: One is a polymer carrier (HPMA), which transports metal organic drugs such as Mn(CO)_3_ light-induced CORMs; the second is a biodegradable polymer drug connector; the third is an anti-tumor agent. The Bruckmann and Kunz groups reported that N-(2-hydroxypropyl)methacrylamide (HPMA) and pentacarbonyl bromide were heated under the re-flux of a dry acetone solution to obtain a copolymer P1 at high yield (Figure 21) [155]. Inducing CORMs can passively transport CO to release metal drugs for tumor cells or sites of inflammation. The polymer conjugate P1 releases CO at 365 nm light, which is not cytotoxic to the HCT116 human colon cancer and HepG2 liver cancer cells.

HPMA-copolymer has a great nature of water solubility. Bernhard Spingler et al. discovered the copolymer materials [156]. The bis(2-pyridylmethyl)benzylamine ligand was prepared from picolyl chloride and benzylamine, then this ligand coordinated with Re(CO)_3_ moiety to construct the HPMA-co-bis(2-pyridylmethyl)-4-vinylbenzylamine copolymers (Figure 22). HPMA-copolymer characteristics such as the average molecular weight can be modified according to the requirement by replacing the radical starter and co-monomers. For instance, the molecular weight 52KDa is the appropriate choice for remarkably enhancing the permeability and retention (EPR) effect. The established copolymer system with the Re(CO)_3_ fragment has an ability to diagnose the used 99mTc. Moreover, the identical behavior in IR spectra and X-ray crystallography have shown their resemblance in the binding sites of Re(CO)_3_-labelled copolymer and bis(2-pyridylmethyl)-amine-derived complexes (solid-state structures (SSS) of ·CH_2_Cl_2_ and·CH_2_Cl_2_·H_2_O). HPMA-morphology can be efficient for targeting the tumor sites for the EPR effect.

Not only solvent exchange triggers complexes have been reported but photons energy has also been explored in this research area. Pierri et al. have shown that the water-soluble photo-CORM can be controlled through NIR photons energy. They utilized the amphiphilic polymer for encapsulating the Photo-CORM *trans*-Mn(bpy)(PPh_3_)(CO_2_)_2_ (Figure 23) [47]. Since CO has a strong ligand field (L.F), so its absorption bands lies in higher energy zone almost closer to the UV-region [162]. Usually Photo-CORMats belongs to the MCCs family and having CO photolysis-lability from the excited states of LF associated metal-centered [163]. Although the higher MLCT transitions obviate the ligand re-configuration, during the MLCT exhibition, the photolysis might be affected due to π-back-bonding of metal-to-CO configuration [164]. The energy gaps were found between the MLCT and LF excited states. The lower excitation states of MCLT have reduced the ligand lability as compared to LF states [163]. Thus, it is difficult to construct the Photo-induced-CO-release upon NIR or longer visible wavelengths. Up-conversion nanoparticles (UCNPs), (i.e., lanthanide ion doped) have a utility of uncaging from NIR photolysis wavelengths. UCNPs are already claimed as photodynamic therapy [165,166]. Those research analysts firstly developed the polymer matrix UNCPs@PL-PEG (an amphiphilic phospholipid-functionalized poly(ethylene glycol) then used NIR photons energy as a trigger to release the CO segment for biological purposes. PL-PEG shows a remarkable characteristic of water solubility and it has also provided space for incoming other soluble photo-organic CORMs. These organic compounds might be helpful for searching the special physiological targets.

### 2.6. Metal Organic Frameworks (MOFs-CORMats)

Metzler-Nolte et al. synthesized the hybrid material metal-organic framework as a class of porous coordinated polymer to encapsulate the CO segment. [138]. They established the bio-compatibility with MOFs; NH_2_-MIL-88B (Fe) and MIL-88B (Fe) by capturing the CO at susceptible Fe^II^ and Fe^III^ coordinative unsaturated metal sites (CUSs). Unfavorably, it requires higher activation temperature [167] (≥ 550 K). An adventuring feature of MOF has a breathing effect while providing accommodation for CO at the adjacent site; probably having a controlled CO release, potential through the opening/closing gesture of porous MOF. 

Instead, it encapsulates the mere CO segment. The research can also accommodate the CORM’s fragment inside MOFs. A new class of Manganese carbonyl complex Photo-CORMs has been explored from Zr(IV) based MOF. MnBr(bpydc)(CO)_3_ (bpydc=5,5’-dicarboxylate-2,2’-bipyridine) embedded into Zr(IV) MOFs [137]. This photo-activated commodity has been evaluated into cellular substrates along with the biocompatible polymer matrix, claimed to be controlled and efficient light-induced CO-release. After the successful CO release from CORMats, the inability in containment of metal degradation growth, this leads to scientists being reluctant for the employment of genuine medication. The CO has played its own role effectively; how other body organisms responds remains questionable.

### 2.7. Porous Structure Materials

Porous structure materials have already been in use for encapsulation of the CORMs commodity for the usage of the CO release. Maldonado, Elisa Barea et al. shared the amazing research work about the organometallic community by embedding the anion porous framework with the exhibition of the cation exchange strategy (Figure 24) [50]. This innovation invigorates under physiological condition while it is triggered by visible light. At first, they developed the air-stable, nontoxic, photoactive and water-soluble cation species CORM [Mn(tacn)(CO)_3_]^+^ (ALF472^+^) then encapsulates into anionic porous matrixes belonging to the inorganic framework. Anionic silica matrix exerts the good administration by reducing the CO release kinetics and providing the control-able rate. The ALF472@hybrid silica-SO_3_ material might support the on/off switch delivery management, probably experiencing the command and control on the released CO. For ensuring the safety of the CORMats, the process metal degradation fragment was examined for ALF472@MCM-41-SO_3_ up to 72 h and no significant appearance was reported. This silica framework is providing an 80% CORMs metal fragment containment. In phototherapies, the CO supplied is a provocative feature in the controlled manner, such as many inflammatory skin issues and topical skin cancer treatment.

Although many of the CORMats have been explored for therapeutic purposes, but the lack of biocompatibility and inconvenient features like solubility and toxicity from organometallic compounds, it could not be used as a drug agent. To overcome this dilemma, CORM-1 has been embedded into the nonporous fibers structure poly(L-lactide-co-D/L-lactide). The bioavailability and water accessibility were confidently achieved by photo-activated electro-spinning [111]. Specifically, there was no toxicity observed during the mouse fibroblast 3T3 cells culture. This feature might be promoting the CORMats into viable drug materials in the future.

### 2.8. Nanaoparticles

Among the CO-carrying support system, nanoparticles are intensely focused because nanoparticles can passively target the malignant cells and actively involve with the targeted tumor cells. Ulrich Schatzschneider et al. open the gateway for the nanoparticles carriers of Silicium dioxide as photoactivatable CORMats [157]. These research analysts, first time synthesized the 3-azidopropane-functionalized SiO_2_ nanoparticles, followed by [Mn(CO)_3_(tpm-L1)] in a dimethylamide solution at room temperature. Manganese-based Photo-CORM [Mn(CO)_3_(tpm)]^+^ whose tpm ligand is linked with Silicium dioxide nanoparticles by CuAAC “click” reaction through construction of the Azido group by emulsion copolymerization of (3-azidopropyl)triethoxysilane and trimethoxymethylsilane at the surface. In this mechanism nanoparticles were functionalized with manganese tricarbonyl as light-induced CORMats (Figure 25). SiO_2_ nanoparticles are stable in nature with an amazing bio-compatibly and easy to modify at the surface. It means that various target molecules could be incorporated at the surface to achieve the drug delivery objectives. Eventually, it will increase the drug delivery capacity at specific sites. So therefore, reduce the systemic risk. To reduce the side effects of chemotherapy, SiO_2_ nanoparticles provide the most valuable platform for tumor drug delivery.

In the nanoparticle carrier strategy, Urara Hasegawa et al. introduced the CO-releasing polymeric nanoparticles (CONPs) through phenylboronic acid-catechol complexation of catechol-bearing CO-donor Ru(CO)_3_Cl(L-DOPA) and phenylboronic acid-containing framboidal nanoparticles. It has been testified in a biological organism and gives feedback to cysteine, and subdues the pro-inflammatory mediator’s IL-6 [168]. 

### 2.9. Nanosheets 

The encapsulation of CORM’s commodity shows promising features. X. Chen et al. worked out to cage the Manganese-carbonyl CORM inside the small MnCO-graphene oxide (PEG-BPY[MnBr(CO)_3_]-GO) nanosheet, recruited as a drug carrier trigger by NIR light energy with on-demand CO release for photochemical CORMats [48]. They successfully constructed the novel combinations of the CO-release mechanism but triggered facility provided inconvenience for therapeutic purposes. However, we also need to pay attention to its complete management system, as in some cases big trouble may be faced for its state of being clinically applied or not. Merely an advantage for the CO releasing behavior is not enough, as it will always be difficult to handle it and nearly impossible for remote area patients, thus it raises a concern for being a practically viable medicine or not. Moreover, there might be no concern to what happens to metal degradation and leftover residue, which actually needs to be addressed properly too. Apart from this, how can we provide this medication to the patient using NIR trigging, so its mobility will remain the utmost challenge. In reality, the novel production in terms of its laboratory scale is quite different from its practical application as a cure agent.

### 2.10. Metallodendrimers

Metallodendrimers has a monodisperse nature with facile preparation. This exclusive feature is demonstrated by Smith et al. to entrepreneur the Photo-CORMats (Figure 26) [161]. For this purpose, Photo-CORM Mn(CO)_3_ moiety scaffolds with polypyridyl dendritic. The general representation of polypyridyl dendritic is [DAB-PPI-{MnBr(bpy^CH3,CH=N^)(CO)_3_}_n_] (whereas DAB=1,4-diaminobutane, PPI=poly(propyleneimine), bpy=bipyridyl). Photo-activated metallodendritic CORMats has been observed to liberate CO molecules upon 410 nm visible light photons penetration. 

### 2.11. Nanodiamond (ND)

The tpm ligand of Photo-CORM [Mn(CO)_3_(tpm)]^+^ peptide material and nanoparticle [Mn(CO)_3_(tpm-L1)] could further be constructed to explore the azide-modified nanodiamond (ND) by CuAAC (copper-catalyzed 1,3-dipolar azide–alkyne cycloaddition) “click” formation as Manganese-MCCs Photo-CORMats (Figure 27) [159]. Dordelmann, G. et al. introduced the first-time CuAAC coupling to attach the CO-liberating agent with ND as a biocompatible supporter. Photoactivatable CORM [Mn(CO)_3_(tpm)]^+^ retained at the ND’s surface for CO biological services and therapeutic purposes and were evaluated through standard myoglobin assay.

Different CORMats are compatible with special cellular environments and are free to perform their therapeutic activities. Certain conditions restrict CORMats activities; definitely it would directly affect the therapeutic performance. CORMats therapeutic potential relies on the material’s nature such as solubility, compatibility and activation mechanism. Another advantage of CORMats is to modify the CORMats assembly according to respective disease cells, which could be more helpful in searching the selective targets. For cancer treatment, the redirected T-cells, i.e., chimeric antigen receptor (CAR) T-cell might be providing a governing principle for cancer therapy [169]. CAR-Tcell is easier to find its own therapeutic targets from the peculiar receptor configuration. This exclusive feature facilitates gene-therapy. Similarly, this morphology can be applied to the CORMats development for special tissue selectivity. Numerous CORMats with their biological significance are described in Table 2.

Briefly if the above intensive research discussion is summarized; it is evident that Ru-MCCs and Mn-MCCs are the right choices for nano-medicine due to bio-compatibility and tremendous prescribed feasibility analysis especially Ru-MCCs due to its lessened toxicity. Scrolling down from micellization to nanodiamond, all CO-prescriptions have CO liberation capability, but none of them could be claimed as safe therapeutic management and can’t be directly applied for exogenous CO-prodrug. During the CORMats administration, rather than focus on the exploration of new advanced materials, the researchers might be considering already existing pharmaceutical materials. A pharmaceutical drug like substance such as crystalline smectite clay; is one of the promising biocompatible and pharmaceutical composites, which could be transformed into CORMats after proper formulation and careful administration. There are two types of strategies that have been introduced for the CORMats production. One is exploiting from the already developed CORMs with biocompatible materials as CO carriers. Another analogy is captured from the CO moiety in the vicinity of material specific akin to MOFs [138]. Already developed CO incorporating strategies are generalized in Table 3.

## 3. CO-Releasing Kinetic Profile 

CORMs and CORMats must have a CO utility to deliver in response to the biological system soon after trigger. The specified trigger plays a decisive role for therapeutic applications. Their kinetics is highly dependent on the trigger facility at which they are applied for. The CO discharging rate was exclusively committed for searching the affected sites of selected targets. The half-life (*t*_1/2_) of CORMs/CORMats (*t*_1/2_ is defined in time duration as half of the introduced CORMs/CORMats amount will be dis-integrated) is the key parameter for examining the CORMs/CORMats stability and sustainability. The fast CO-releasing rate is difficult to attain predetermined clinical objectives. 

Just an illustration [170], the half-life (*t*_1/2_) of CORM-3 is 3.6 min only when anticipated with the human plasma. At that moment, CORM-3 dissolves in plasma configuration and suddenly reacting with albumin, supplies CO_2_ and Ru(CO)_2_ segment; and also makes an alliance with protein in vivo circulation, where CO serves slowly and nonspecifically [171]. Non-technical CO release is unable to deliver the necessary pharmaceutical features. Likewise slow and fast CO release molecules would be engineered to accommodate the distinct clinical trials (Figure 28).

The half-life of CORM-1 and CORM-2 is about 1 min in PBS (*phosphate buffered saline*) at 37 temperature with pH~7.4 [37,172]. Such types of half-lives are considered very short intervals. The CORMs and CORMats deliberation must be regulated along with integral body fluids in order to communicate with victim organs and/or tissues before CORMs/CORMats (as CO-producer) consume entire CO quantity [173]. To improve the sustainability of CO carriers, the half-life (*t*_1/2_) should be extended for few minutes, but somehow few seconds and milliseconds extension will be more beneficial. A different designated strategy has promoted the transient CO releases. These mechanisms will be observed through ion-channel kinetic studies [174]. The extended pharmacokinetic qualities containing nanomaterials (NPs) and macromolecular models could be exploited for the management of CO transporters or CO carriers.

## 4. CORMs/CORMats Cytotoxicity and Tissue Accumulation

In addition to CORMs and CORMats pharmaceutical advantages, it delivers some adverse effects too because of their toxicological profile or even proliferation of toxic metal residues (i-CORMs/i-CORMats) resulting soon after CORMs/CORMats launch the CO into the biological environment [95,175,176,177]. Subsequently, a particular deficiency of CORMs/CORMats is usually observed after the CO excretion; their CO-missing analogues tend to prevail in situ administration. Therefore the transition-heavy metal core usually harbors cofactors and is involved in some uncontrolled reactions/activities with neighboring tissues/cells, thereby contributing serious cellular impairment (Figure 29).

Wang et al. studied the in vivo toxicity, cytotoxicity, metabolism and bio-distribution of two carbonyl metal CORMs series including Ru(CO)_3_ClnL and M(CO)_5_L (M= Cr, Mo, W) [178]. The cytotoxic effect was monitored on murine macrophages through MTT colorimetric assay with respect to IC_50_ and LD_50_ values; the severely damaged kidney and liver were observed to picture both morphological and functional aspects. The cell culture RAW264.7 was incubated with CORMs/CORMats while examining cytotoxicity and demonstrates their bactericidal activity against a variety of microbes, including *Staphylococcus aureus, Escherichia coli, and Pseudomonas aeruginosa* [126,127]. In this study, they found the uneven distribution of metal complexes in organs and tissues that subsequent damage through metal ions oxidation such as Ruthenium complexes. It was oxidized from Ru^II^ to Ru^III^ by P450 enzymes. This toxicity issue was elucidated by Winburn et al., they also performed the CORM’s toxicological profile [179]. CORM-2 and its depleted form i-CORM-2 were studied and compared in two kidney cell lines (MDCK and HeK lines) and primary rat cardiomyocytes. This study explained that the CORM-2 cytoprotective concentration (<20mM) is approaching to cytotoxic value (>100 mM). Moreover, both CORM-2 and i-CORM-2 exerted the cellular toxicity by means of the abnormal cell cytology, cell cycle arrest, reduced cell viability, increased apoptosis and inhibited mitochondrial enzyme activity [136]. These particular consequences were observed through the metal-core mediated toxicity. Different studies have also been explored that intensifying the polarity of CORMs/CORMats would be possibly limiting their penetration over the cellular membrane, and thus attenuating their toxicity [136].

## 5. Concluding Remarks

CO is generally infamous for its toxicity, but a controlled dose of CO shows useful biological impacts. The CO’s detail analysis exhibits endogenous production by heme oxygenase and explores the therapeutic scope. This scientific study not only confirms the endogenous generation of CO, which has important potential in pathological tissues but also guarantees exogenously released CO’s therapeutic impacts. Therefore, the challenges for the pharmaceutical drug chemists have always been and are continuing still, for the development of a risk-free and more convenient strategy to deliver therapeutic CO dosage. The CO administration with biological system suggests their therapeutic potential. This CO administration relies on MCCs for CO liberation. Thus, CORMats were developed by MCCs with different conjugate/scaffold systems. The CORMats have been covalently assembled with different nanomaterial including polymers, silica nanoparticles, proteins cages, vitamins, metallodendrimer, micelles, nanodiamond and nanofiber gel (peptide amphiphilic) or even incorporated with magnetic nanoparticles (maghemite), tablets, non-woven, or either MOFs for following features: To enhance sustainability and stability; to approach the special cellular tissues/organs; to reduce the toxicity; to attain the EPR- effect; or to permit special triggers facilities. CORM has the capability to deliver the CO to tissues and cells in vivo, in-fact constitute the most appropriate scheme to accomplish the therapeutic outcomes. This proof-of-concept refers to the medicinal chemists to endeavor modern CORMats furnished with ADME (CORMats characteristics: Administration, Distribution, Metabolism, Excretion), prerequisite for the clinical utility. As a prodrug, these developed mechanisms are highly dependent on in vivo performance. It has been worth mentioning that many pharmaceutical materials were also claimed to be non-toxic such as smectite clay that might be transformed into CORMats for promising therapeutic benefits. Probably crystalline smectite clays are the best choice for the CORMats development due to its con-comitant administration. Additionally, their layered structure exfoliations and cation exchange capacity (CEC) have been encouraging for developing the new class of CORMats. Furthermore, it is mandatory to investigate the metal residues (remaining fragments) after CO liberation, if any side effect of newly developed materials is reported should try to minimize it by modern carrier designs. The aim of the controlled CO delivery management was sponsored by tissue selection and distribution. The CORMats activation with different triggers did not permit to develop “universal” CORMats for every disease model. The method of CORMs/CORMats trigger or even CO activation is used to disintegrate the MCCs through photo, thermal, enzyme, pH, oxidation and solvent trigger CORMs/CORMats bearing ligand exchange strategies. These CORMs/CORMats strategies are promising candidates of the therapeutic potential and deserve exclusive attention for thorough therapeutic investigations.

The toxicity of the CO precursor is still a big challenge for the researchers. The CORMs/CORMats toxicity was in-action during and after the CO release with depleted metal residues abbreviated as i-CORMs/CORMats. The fast CO release helps to study the ion-channel path, while the slow release favors tissue targets. It is mandatory that CORMs/i-CORMs and CORMats/i-CORMats (before/after CO release) should not be participating in any toxic activity. Otherwise it will not be possible to prescribe for patients; as the safety of human organs is the utmost priority. 

The above discussion confirms that CORMs and CORMats are accountable for the CO-produce being the active ingredient. It should be noted that CORMats did not technically modify specific receptors but only provided a transport and discharge services for the CO gas. Therefore, the therapeutic impacts of CORMats under physiological conditions to employ CO preferentially and professionally against damaged biological tissues/organism must prevail and ensure the quick release of loaded CO upon trigger.

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
