# Peer review of "CO-Releasing Materials: An Emphasis on Therapeutic Implications, as Release and Subsequent Cytotoxicity Are the Part of Therapy"

_materials, 2019, doi:10.3390/ma12101643_

Reviewer 1 Report

The presented review describes and analyzes the applications of CO-releasing materials and associated toxicity.

In general, the review and topic are interesting and timely.

However, the language used is very poor quality. It starts from the very beginning of the article, including the Title.

I strongly recommend the editing and revising the manuscript with the help of an English native speaker.

And I also suggest sub-dividing/splitting Chapter 1 and Chapter 2 into smaller subsections and re-structure the whole article.

Secondly, I advise the inclusion of fresh articles published within recent 5 years, because the majority of referenced publications is outmoded.

Author Response

Dear Reviewer

Complete Answer is given in attached file. Please Conisder it. 

Kind Regards

Author

Reviewer 2 Report

The article presents the fabrication of CO-releasing materials for biomedical applications. Initially, the report presented the coagulation and fibrinolysis scope of CORMATs and then on cytotoxicity as well as releasing profiles. The article was well focused; however, numerous grammatical errors made it difficult to read. I suggest publishing after an ample amount of revision.

The introduction needs to be rearranged as no discussion is highlighted on what materials are exactly CORMats and their advantages until the end. I suggest explaining the materials used and also move the benefits of CO molecules from the introduction by incorporating them as a separate section.

Any scope for initiatives on clinical translation of such materials reported yet?

The title is in a way too long, better concise as “CO-releasing Materials: An emphasis on therapeutic implications” as release and subsequent cytotoxicity are part of therapy

Need proofreading by a native speaker, as numerous language errors and typo errors are evidenced.

Minor

Abbreviations are irregularly presented, better unify for instance CORMats and CO-RMats.

Abbreviations should be defined when they appear for the first time

Change heading 2 as Research on new CO transport materials.

Author Response

Dear Reviewer

Complete answer of your comments and revised manucript is attached in file. 

Kind Regards

Author

Round  2

Reviewer 1 Report

The article has been significantly improved in terms of language and construction.

But I have found out some minor grammatical errors, so please revise the text again.

However, I do recommend to accept this article for the publication if grammatical problems will be fixed.

Author Response

Dear Reviewers

 I checked this manucript by the english nationality holder, And removes the grammar mistakes.

This new version is uploaded in .DOCX format in attached file. 

Regards

Author
